# Embrittlement Mechanisms of HR3C Pipe Steel at Room Temperature in Ultra-Supercritical Unit

**DOI:** 10.3390/nano14030306

**Published:** 2024-02-02

**Authors:** Xinying Liu, Xue Cao, Zhongwu Zhang

**Affiliations:** 1Key Laboratory of Superlight Materials and Surface Technology, Ministry of Education, College of Materials Science and Chemical Engineering, Harbin Engineering University, Harbin 150001, China; liuxinying@hrbeu.edu.cn; 2Shandong Branch of Huadian Electric Research Institute, Jinan 250014, China; 3College of Computer Science and Technology, Harbin Engineering University, Harbin 150001, China

**Keywords:** HR3C steel, M_23_C_6_, σ phases, ultra-supercritical boiler, embrittlement mechanism

## Abstract

HR3C steel is an austenitic high-temperature-resistant steel. Because of its good strength and high-temperature performance, it has been widely used in ultra-supercritical power plant boilers. With the increasingly frequent start-up and shutdown of thermal power units, leakages of HR3C steel pipes have occasionally occurred due to the embrittlement of HR3C pipe steel after a long service duration. In this study, the embrittlement mechanisms of HR3C pipe steel are investigated systematically. The mechanical properties of the pipe steel after running for 70,000 h in an ultra-supercritical unit were determined. As a comparison, the pipe steel supplied in the same batch was aged at 700 degrees Celsius for 500 h. The mechanical properties and the micro-precipitation of the aged counterparts were also determined for comparison. The results show that the embrittlement of HR3C pipe steel in service for 70,000 h is obvious. The average impact absorption is only 5.5 J, which is a decrease of 96.7%. It is found that embrittlement of HR3C steel also occurs after 500 h of aging at 700 °C, and the average value of impact absorption energy decreases by 70.4%. The comparison experiment between the in-service pipe steel and the aged pipe steel shows that in the rapid decline stage of the impact toughness of HR3C steel, the M_23_C_6_ carbide in the microstructure has a continuous chain distribution in the grain boundary. There were no other precipitated phases observed. The rapid precipitation and aggregation of M_23_C_6_ carbides leads to the initial embrittlement of HR3C steel at room temperature. The CRFe-type σ phase was found in the transmission electron microscope (TEM) image of the steel pipe after 70 thousand hours of use. The precipitation of the σ phase further induces the embrittlement of HR3C pipe steel after a long service duration.

## 1. Introduction

Ultra-supercritical units are currently the most efficient coal-fired generating units because of their high steam pressure and temperature [1,2,3]. HR3C steel is a kind of pipe steel obtained by adding Nb and N elements to 25Cr-20Ni austenitic steel from Sumitomo Corporation of Tokyo, Japan, and is widely used in ultra-supercritical power plant boilers due to its high-temperature strength, oxidation resistance and corrosion resistance [4,5,6,7]. Between 2021 and 2023, with the increasingly frequent start-up and shutdown of thermal power units and depth peaking, the number of HR3C steel pipe leakages has gradually increased. The embrittlement of HR3C pipe steel after a long service duration is considered to the reason for the failure and leakage of the pipes [2,8,9].

It is reported that the formation and accumulation of precipitates in the high-temperature service process lead to a decrease in impact energy and to embrittlement [10,11,12,13,14,15]. In general, the content of C is limited to a low level in HR3C steel. The formation and the sequence of various precipitates play an important role in inducing embrittlement. However, the process of precipitation during service lasts for a long time and it is hard to trace the formation of precipitates. In this study, the mechanical properties of pipe steel after running for 70,000 h in an ultra-supercritical unit were determined. In addition, using the same batch of supplied HR3C steel pipes, a group of pipes were aged for 500 h at 700 degrees Celsius. The mechanical properties and precipitation of these three kinds of steels were characterized. The formation of various precipitates was investigated. These studies may lead to a deeper understanding of the formation of precipitates during service and provide a basis for further engineering replacements and service life evaluation of this kind of heat-resistant pipe steel.

## 2. Materials and Methods

The HR3C steel pipe studied in this paper was imported by Sumitomo Corporation of Japan, and the service steel pipe was taken from the outlet pipe section of the high-temperature superheater of Unit 1 of Huadian Laizhou Power Plant, with a specification of 51 × 11.4 mm. This steel pipe has been in service for about 70,000 h at present. The supply pipe used in this experiment was the spare pipe of the same batch as the steel pipe in service. The same batch of spare steel pipe was used in this experiment, which can exclude the influence of metallurgical manufacturing state on the experimental results. The element content of the steel pipe is shown in Table 1.

Although the alloying elements of HR3C steel and their composition range are specified in the ASME standard, in fact, there are more stringent requirements for their composition range during the development process. The restrictions on C, P and S are mainly to facilitate the welding of materials; Si and Mn are elements that promote the formation of the σ phase, and their proper restriction inhibits the precipitation of the σ phase. When Nb is in the upper middle limit, the combination with N, C and Cr can have a more obvious effect on strengthening precipitation and aging. For Ni and N, it has been proved that when the mass fraction of Ni and N is lower than the lower limit, the normal phase will be precipitated during the long-term aging process. When the mass fraction of Ni and N is higher than the upper limit, in addition to the precipitation of NB-rich carbon nitride, the Cr_2_N and π phases will be precipitated, which will reduce the toughness of the material. N and C are the same interstitial elements; increasing the mass fraction of N in steel can improve the high-temperature strength of the material, stabilize the austenitic phase and improve resistance to intergranular corrosion and point corrosion.

The supply HR3C steel pipe was subjected to an aging test at 700 °C; the aging time was 500 h, and air cooling was performed after aging. Three kinds of HR3C steel pipes were used for comparative experiments: steel pipes in the supply state, steel pipes after aging and steel pipes after 70,000 h of use. The samples were made according to the experimental standard specifications, and the metallographic and room-temperature mechanical properties were tested. The metallographic microstructure was observed under a DMI5000M optical microscope (Leica Microsystems, Wetzlar, Germany) with ferric chloride hydrochloric water solution (with a ratio of 5 g of ferric chloride, 50 mL of concentrated hydrochloric acid and 100 mL of water) as the etchant. Tensile and impact tests were performed using a CMT5605 electronic universal testing machine (Sansitest, Shenzhen, China) and ZBC3302-A pendulum impact testing machine (Zhejiang Bangcheng Testing Machine Co., Ltd., Hangzhou, China) according to ASTM A370-2017 [16]. The impact fracture morphology and cross-section analysis were observed using Zeiss EVO 18 internal energy spectrum scanning electron microscopy (ZEISS, Tokyo, Japan). A sample of 10 mm × 10 mm × 0.5 mm was cut and thinned to 80 μm by sandpaper, a TEM sample of 3 mm was pressed out, and then the TEM sample was thinned by double-spraying in 8 vol% perchloric acid alcohol (temperature about −25 °C) for TEM characterization. The transmission electron microscope model used was FEI Talos F200X (Thermo Fisher Scientific, Waltham, MA, USA).

## 3. Experimental Results

### 3.1. Metallographic Test

The metallographic structure of the HR3C steel pipe is austenite (Figure 1). There are obvious annealing twins and part of the undissolved phase in the grain of the supplied steel pipe, and the grain boundary is not obvious. In the metallographic structure of the steel pipe after aging, some unmelted phases disappear, and grain boundaries begin to appear with discontinuous massive carbides. The austenite structure of the service steel pipe is unchanged, a large amount of precipitates accumulate near the grain boundary, there are precipitates in the grain, the twins disappear, and the grain boundary is clearly shown due to the accumulation of a large amount of precipitates.

### 3.2. Tensile Test

The representative tensile strain curves and fracture morphology of three HR3C steel tube samples at room temperature are shown in Figure 2. In the ASME SA213/SA213M standard [17], the yield strength of HR3C steel is required to be not less than 295 MPa and the ultimate tensile strength not less than 655 MPa, with an elongation after breaking of not less than 30%. The yield strength and ultimate tensile strength of all samples in this experiment were higher than the requirements of the ASME standard. The yield strength of the supplied steel pipe was 481 MPa, the ultimate tensile strength was 777 MPa and the elongation at break was 31%. After aging, the yield strength of the steel pipe was 502 MPa, the ultimate tensile strength was 716 MPa and the elongation at break was 10%. After service, the yield strength of the steel pipe was 437 MPa, the ultimate tensile strength was 765 MPa and the elongation at break was 5%. The fracture of the supply state showed obvious plastic deformation at the macro level, the dimple size was relatively uniform at the micro level and the morphology was small-scale dimple, which is a typical ductile fracture mechanism. The brittle fracture morphology of the steel pipe after aging and service was observed using a scanning electron microscope. The crystal sugar-like grain was typical of intergranular brittle cracking, and there were obvious intergranular secondary cracks, but after aging, the surface of the steel pipe section grain was more rough.

The ultimate tensile resistance of HR3C steel is one of the mechanical properties that need to be tracked during its service. HR3C steel is an austenitic heat-resistant steel, and the most important strengthening method used is the relatively singular precipitation phase strengthening; martensitic heat-resistant steel does not possess strip strengthening, dislocation strengthening or other strengthening mechanisms [18]. Therefore, a large amount of second-phase precipitation during service is the key reason for the change in the impact work and tensile properties of HR3C steel. With the gradual increase in HR3C high-temperature service time, the tensile strength decreases slightly at first and then increases, and the overall tensile strength remains high. With the increase in service time, the yield strength of HR3C steel samples generally decreased, but the decreasing range was limited, and our experiment proves that the use of 70,000 h service time is still higher than the standard requirements. However, with an extension in service time, the elongation of the alloy after breaking decreased significantly. Further research results show that the precipitated nano-phase at high temperature is the reason why HR3C has excellent tensile properties at high temperature [3,19,20,21].

At present, less attention has been paid to the variation in tensile properties of HR3C steel after long-term high-temperature service, but it is an important indicator to ensure the safe use of the material after long-term service, and it is important to track and study its changes.

### 3.3. Impact Test

The impact energy absorption of HR3C steel pipe at room temperature changes greatly after high-temperature aging or service. The average impact energy absorption at room temperature of the supply steel pipe was 169 J, the average impact energy absorption at room temperature of the aging steel pipe was 50 J, and the average impact energy absorption at room temperature of the service steel pipe was about 5.5 J, which was 96.7% lower than that of the supply steel pipe. The impact fracture of the HR3C supply steel pipe at room temperature showed obvious plastic deformation at the macro scale, and the enlarged morphology shows small-scale dimples, as shown in Figure 3a, which is a typical ductile fracture morphology. The HR3C service steel pipe showed no plastic deformation at the macro level, and the microstructure is shown in Figure 3c. Like a large crystalline sugar cube, there are many flat small sections, which are clear without dimples, and there are brittle fractures along the crystal and secondary cracks between the crystals [22].

### 3.4. Result

The mechanical properties of the HR3C steel pipe at room temperature after 500 h of aging at 700 degrees Celsius show that the elongation after fracture was significantly reduced, the fracture presented a brittle fracture characteristic, and the average value of impact absorption work was reduced by 70.4%, indicating an embrittlement phenomenon. After 70,000 h of service, the embrittlement degree of HR3C steel pipe was further aggravated, and the average value of impact absorption work was only 5.5 J, a decrease of 96.7%.

Due to the austenite precipitation strengthening of HR3C steel, the ultimate tensile strength and yield strength of all samples at room temperature were higher than the standard requirements.

## 4. Discussion

Through the above experiments, it was found that the HR3C steel pipe after aging at room temperature and after service had different degrees of embrittlement phenomenon, the impact absorption energy was reduced, and the fracture cracked along the grain boundary. The reason for embrittlement is twofold. First, when the impurity elements S and P occupy the grain boundary position, more charge is transferred to the impurity atoms during the bonding process, and the bond cooperation of the standard atoms is weakened, so that the grain boundary binding force is reduced and the brittleness is increased [23,24]. Second, the continuous and flaky precipitation of the carbide M_23_C_6_ at the grain boundary reduces the cooperative deformation ability between grains, making it easy for dislocation to accumulate at the grain boundary and increase the brittleness of the material [25,26].

### 4.1. Segregation of Elements P and S

The distribution of the elements S and P in the grain boundaries of the three samples is shown in Figure 4. The segregation of P and S elements did not occur in the supplied steel pipe and the steel pipe after aging. After 70 thousand hours of service, P and S elements showed a segregation phenomenon, and the segregation of S elements was more obvious. However, embrittlement occurred after aging, so the segregation of P and S elements was not the cause of early embrittlement.

### 4.2. Evolution of Precipitated Phase of HR3C Steel

The microstructure of the old and new HR3C steel pipes and the changes in the strengthened precipitation phase were characterized by three characteristics. The first feature is that the twin substructure of austenite grain basically disappeared after high temperature service for 70,000 h. The second feature is that the austenite grain boundaries became obvious and the grain boundary width widened. The third feature is the formation, growth and evolution of precipitates inside the grain; the number of precipitates randomly distributed inside the grain increased slightly after service, and the size increased significantly [27,28,29].

Figure 5 shows the TEM and electron diffraction images of twins and NbCrN precipitates in the impact fracture sample of the service steel pipe. Figure 5a shows the grain morphology of the impact fracture specimen. Figure 5b,c are TEM images and electron diffraction patterns of the twin. The diffraction electrons of the HR3C twin exhibit the following three characteristics: first, the diffraction spots are neatly arranged on the electron layer line; second, diffraction spots appear individually in the 0th layer; third, the diffraction spots of the ±1 layer and ±2 layer appear in pairs [30]. Figure 5d–f show that polygonal particles with a length of about 300–400 nm were distributed in the crystal, and the electron diffraction pattern and EDS experiment results showed that they were NbCrN nitrides with simple tetragonal structure. The results of EDS quality scores are shown in Table 2. NbCrN has a significant hardening effect relative to the properties of austenitic steel. The fine NbCrN particles produce orthopedic dislocations and increase the hardness, thus improving the creep properties of HR3C steel. However, with the coarsening of the NbCrN phase, its ability to resist dislocation is weakened.

The twin boundary in the grain was divided into a coherent twin and non-coherent twin, which could easily be discerned in the solution treatment. This can release the compressive stress around the austenitic grain boundary and destroy the continuity of the grain boundary, so as to improve the plasticity and toughness of HR3C steel and reduce the rate of crack propagation through the austenitic grain boundary [27,31]. The twinning of the supplied steel pipe and the aged steel pipe showed little change, but the elongation and section morphology of the aged steel pipe after fracture show that embrittlement occurred, which indicates that the initial embrittlement of HR3C has little connection with the twin.

Compared with the original supply steel pipe, the quantity and size of the NbCrN phase randomly distributed in the grain of the aged steel pipe did not increase significantly, but the aged steel pipe was embrittled, indicating that the early embrittlement of HR3C has little connection with the NbCrN phase. Moreover, the NbCrN phase can enhance the hardness and strength of austenitic steel, producing pinning dislocation which results in increased hardness, which will improve the fracture resistance of HR3C steel [32,33].

The grain boundary of the aged steel pipe became obvious because of the presence of carbides. TEM images showed that the precipitated carbide particles had a broken chain shape and were distributed along the grain boundary, with some of the long strips of carbide parallel to each other. EDS analysis results showed the presence of Cr_16_Fe_5_Ni_2_C_6_ (M_23_C_6_) carbides, the longest of which was more than 100 nm long (Figure 6). It was found that M_23_C_6_ carbides appear rapidly in the early stage of aging, and usually preferentially precipitate and start to precipitate at the position of carbon atoms and vacancy on the grain boundary or twin boundary, as well as in the area with high dislocation density and distortion energy [27,34]. The size and quantity of M_23_C_6_ carbides increase rapidly with the rapid diffusion of carbon atoms through dislocation and vacancy at grain boundaries over time. When subjected to external forces, the interface between the coarse M_23_C_6_ and the matrix can preferentially become the location of micro-crack nucleation, and the carbide M_23_C_6_ is easily stripped from the matrix, but still close to the grain boundary, so the grain boundary widens significantly with the extension of high-temperature aging time. However, the coarse M_23_C_6_ carbides precipitated along grain boundaries in chain form will seriously weaken the creep strength of the material. Therefore, HR3C steel pipe shows an obvious embrittlement phenomenon in the early stage of service at high temperature, and the impact fractures of embrittlement samples are all intergranular cracking. So, the continuous flaky precipitation of M_23_C_6_ carbide at the grain boundary causes the early embrittlement of HR3C steel. Yang et al. [35] observed through transmission electron microscopy that when the creep time was extended from 1608 h to 17,115 h at 650 °C, the size of M_23_C_6_ carbides increased by about 57%, but increased by about 3% after 27,943 h, and continued to prolong the creep time. The mean diameter of M_23_C_6_ carbides barely changed.

The results show that the continuous and flaky precipitation of M_23_C_6_ carbide at the grain boundary reduces the co-deformation ability between grains, makes it easy for dislocation to accumulate at the grain boundary, and increases the brittleness of the material [22,36]. At the initial stage of creep, fine rod M_23_C_6_ precipitates along the grain boundary, which can inhibit the grain boundary dislocation slip and enhance the fracture resistance of the steel, while large and continuous M_23_C_6_ carbides precipitating and accumulating near the grain boundary in a chain form will seriously weaken the creep strength of the material [37,38]. On the other hand, the precipitation and growth of M_23_C_6_ exhausts the Cr in the matrix, and the Cr content near the grain boundary is relatively reduced, forming a Cr-poor zone, which can easily produce intergranular corrosion for a long time [39].

During the TEM experiment, the regular square precipitates were found in the austenite crystal of the steel pipe with a size of about 50~100 nm, which was identified as the CrFe type σ phase by microenergy spectrum analysis under transmission electron microscopy, and the Cr content was as high as 50% (Figure 7). The σ phase is an intermetallic compound precipitated in the matrix during long-term service, and the temperature range of σ phase precipitation from the γ austenite matrix is generally 600~980 °C, which is consistent with the operating temperature of the experimental service steel pipe. The literature [32,33,40] indicates that the pre-precipitation of M_23_C_6_ at grain boundaries promotes in situ nucleation of the σ phase on M_23_C_6_, and it has been reported that a part of M23C6 will be transformed into the σ phase after long-term aging at 650~850 °C, so the σ phase is precipitated late in the matrix.

Because the σ phase is a brittle phase with high hardness and low plasticity, an increase in the precipitation amount of the σ phase makes the impact absorption work of HR3C steel pipes with 70,000 h of service show a lower result, and the degree of embrittlement is further aggravated. The fracture observation results of the impact sample further show that the fracture form of HR3C steel after long-term aging is intergranular brittle fracture.

## 5. Conclusions

In this paper, the performance test and microscopic analysis were carried out on an HR3C sample pipe after actual operation in a domestic ultra-supercritical power plant, a steel pipe under the same batch supply state, and a steel pipe aged at 700 °C for 500 h, and the following conclusions were obtained:

After 500 h of high-temperature aging, the elongation of HR3C steel pipe after fracture is 10%, far lower than the ASME standard requirements of no less than 30%, and the average impact absorption work was 50 J, 70.4% lower than the supply steel pipe. These data prove that HR3C steel pipe demonstrates brittle changes in the early stage of high-temperature aging.The fracture position of the HR3C steel pipe sample was irregular in the tensile test, and the yield strength and ultimate tensile strength of the three groups of samples were higher than the requirements of ASME standards. Moreover, the tensile data of the steel pipe after high-temperature aging and service for 70,000 h were not lower than that of the steel pipe in the supply state, so HR3C steel pipe has excellent tensile properties.In HR3C steel, continuous chain-cut M_23_C_6_ phases distributed along the grain boundaries were precipitated. P and S elements showed no obvious segregation in the early stage of high-temperature aging, and S elements showed a more obvious segregation phenomenon after service for 70,000 h. This is consistent with the results of Luo’s 40,000 h high-temperature aging study [25].The main reason for the brittleness of HR3C steel at room temperature is the continuous precipitation and aggregation of M23C6 carbides at the grain boundary, because brittle changes have been shown after 500 h of high-temperature aging, but there is no σ phase and other precipitates in the crystal at that time, and P and S elements do not occur with segregation. Compared with the steel pipe that had been in service for 70,000 h, it was shown that a small amount of the σ phase was produced in the crystal after the HR3C steel pipe had been in service for a long time at high temperature. Because the σ phase is a brittle phase with high hardness and low plasticity, the embrittlement degree of HR3C steel pipe is further aggravated.

The high content of alloying elements causes microstructural changes such as phase precipitation and growth of HR3C steel in the process of high-temperature and high-pressure use, which damages its performance. Therefore, the mechanism of microstructure evolution and the law of performance change of HR3C steel during long-term operation need to be further studied to explore anti-embrittlement methods to prevent the generation of M_23_C_6_ and σ phase, or refine M_23_C_6_, reduce the occurrence of power station leakage caused by brittleness and ensure the safe and stable operation of thermal power plant units.

## Figures and Tables

**Figure 1 nanomaterials-14-00306-f001:**
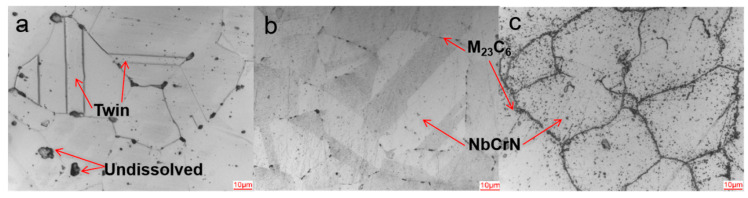
The metallographic organization of HR3C steel pipe; (**a**–**c**) are, respectively, the supply pipe, aged pipe and service pipe.

**Figure 2 nanomaterials-14-00306-f002:**
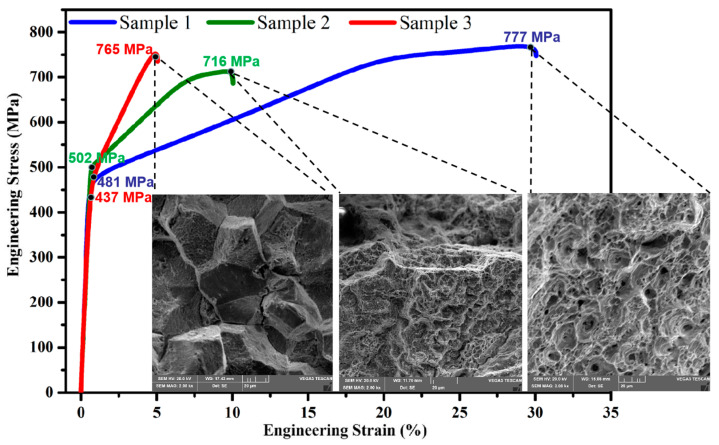
Mechanical properties and fracture morphology. Three typical tensile strain curves at room temperature and corresponding fracture patterns. Sample 1 is the steel pipe in the supply condition, sample 2 is the steel pipe after aging, and sample 3 is the steel pipe after serving 70,000 h.

**Figure 3 nanomaterials-14-00306-f003:**
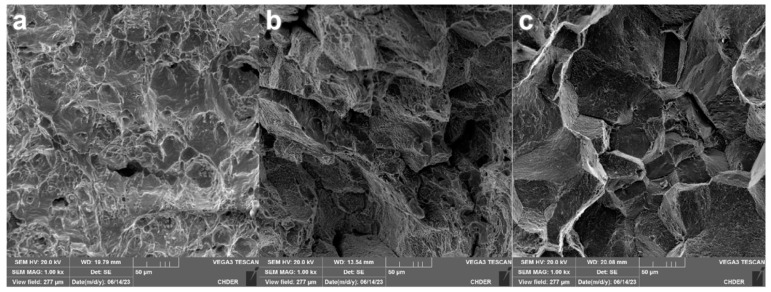
Representative impact fracture morphology; (**a**–**c**) are the supply pipe, aged pipe and service pipe, respectively.

**Figure 4 nanomaterials-14-00306-f004:**
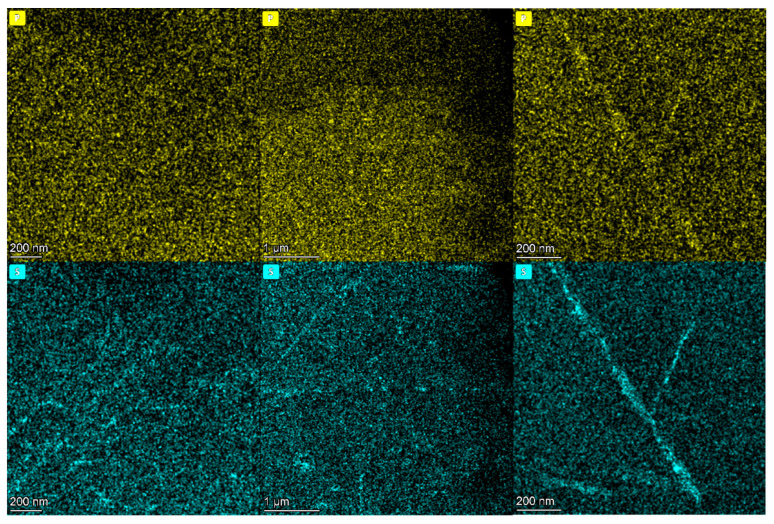
The concentration changes in P and S elements.

**Figure 5 nanomaterials-14-00306-f005:**
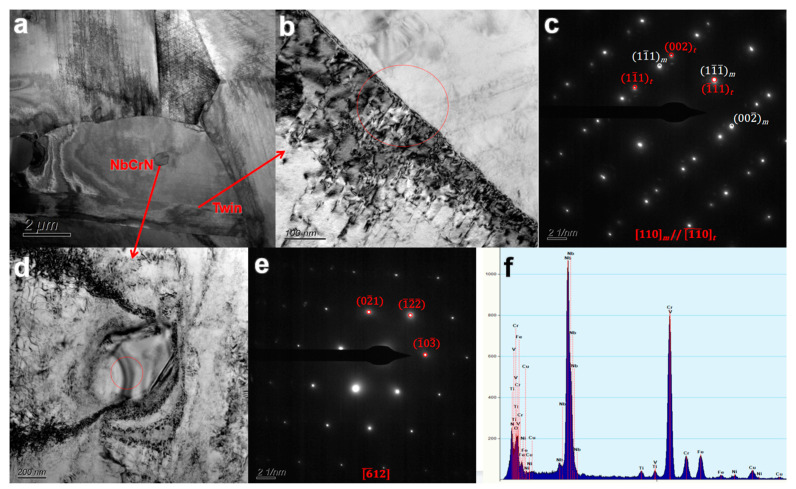
TEM results of twinning and NbCrN precipitates in HR3C. (**a**) is the overall morphology of the grain, (**b**,**c**) is the TEM image and electron diffraction pattern of the twin. (**d**–**f**) is the result of the morphology, diffraction pattern and EDS of NbCrN phase.

**Figure 6 nanomaterials-14-00306-f006:**
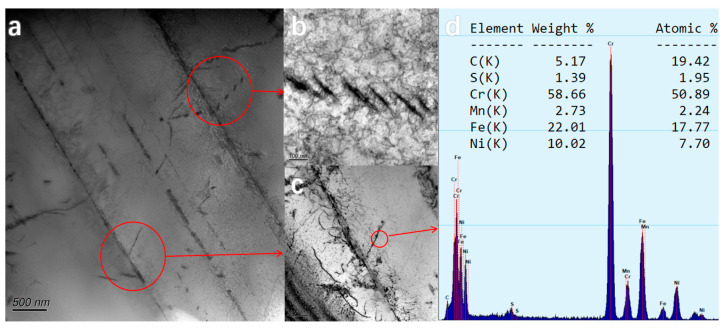
Morphology and EDS spectra of M_23_C_6_ carbides. (**a**–**c**) is the appearance of carbide, (**d**) is the EDS result of carbide.

**Figure 7 nanomaterials-14-00306-f007:**
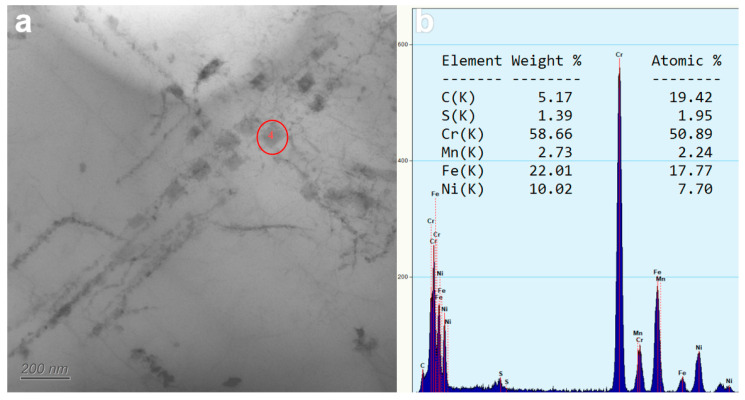
After 70 thousand hours of service, σ-phase precipitation in steel pipes. (**a**) is the appearance of precipitated phase, (**b**) is the EDS result of precipitated phase.

**Table 1 nanomaterials-14-00306-t001:** Elemental analysis results of HR3C steel pipe (wt%).

	C	Mn	Si	Cr	Ni	Nb	N	P	S
Supply steel pipe	0.05	1.16	0.42	25.37	20.97	0.36	0.25	0.021	0.002
Service steel pipe	0.05	1.11	0.43	25.52	20.73	0.42	0.21	0.023	0.002
ASTM	0.04~0.10	≤2.00	≤1.00	24.0~26.0	19.0~22.0	0.20~0.60	0.15~0.35	<0.04	<0.03

**Table 2 nanomaterials-14-00306-t002:** EDS results of NbCrN phase.

Element	Cr	Nb	N	Fe	O	Ti	V	Cu	Ni
Weight %	34.26	28.44	18.07	5.37	7.08	1.36	2.03	2.46	0.89
Atomic %	22.59	10.49	44.23	3.29	15.18	0.97	1.37	1.33	0.52

## Data Availability

The data that support the findings of this study are available from the corresponding authors upon reasonable request.

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
