# Peer review of "Embrittlement Mechanisms of HR3C Pipe Steel at Room Temperature in Ultra-Supercritical Unit"

_nanomaterials, 2024, doi:10.3390/nano14030306_

Round 1
Reviewer 1 Report
Comments and Suggestions for Authors
The paper is of interest, in that it compares properties of as received, HR3C steel, as well as the same alloy subjected to 700C for 500 hrs and 70,000 hrs. The methods description is adequate, but more detail is needed for the microscopy, include specimen preparation. The correspondance between speicmens in fig 1 and fig 2 also needs to be more clear (a, b, c, versus sample 1, 2, 3). In the discussion, p5, line 172 the word "crystal" should be "grain boundary". The discussion of results is worthwhile, but the conclusions are really not clear and don't properly support the results. Please make clear which specimens are being discussed in the conclusions; this is not clear, especially for conclusions 1 and 3.
Comments on the Quality of English LanguageEnglish needs significant revision by native speaker.
Author Response
Many thanks for the valuable comments and suggestions of the reviewer. According to the reviewer’ instructions, we have made corresponding revisions highlighted in blue over the whole manuscript to clarify the points of the referees. Also, please find our responses to the reviewer’ comments point by point below. Please see the attachment.

Reviewer 2 Report
Comments and Suggestions for Authors
Congratulations on the submitted work. The article shows good knowledge by the authors in the area. However, some doubts about your work can be solved by improving the work and their understanding. Please see the attachment.

Author Response

(The authors gave the same response as above.)

Reviewer 3 Report
Comments and Suggestions for Authors
Dear authors, your work has publication potential, but in its current form the article is not suitable for publication. After thorough revisions and reassessment, it may be possible to publish it. My comments are below.
1. on page 2, lines 70 and 71. The authors write about "normal phase", what does it mean and are there also unnormal phases?
2. Why was the aging time of 500 hours used? What was the result?
3. Lack of information on the preparation of thin foils in the research methodology.
4. How did the primary precipitates disappear in the material after aging (page 3, line 96)?
5. Fig. 2 there is no marking in this drawing what sample 1, sample 2, and sample 3 mean. In the same drawing it should be MPa and it is Mpa.
6. Authors on page 5 in point 4.1. discuss the influence of P and S on the brittleness of the analyzed alloy, but do not include the content of these admixtures in Table 1. What was their content in the tested steels? and what was the accuracy of the research method (device) on which these tests were carried out?
7. Page 5, lines 170 - 171 - The authors write about aging at room temperature?
8. Are you sure there are S and P precipitation in Fig. 4? As far as I know, these are changes in the concentration of S and P in a given area!
9. Page 7, line 220, how do the authors know that the amount of NbCrN precipitation did not increase significantly?
10. Isn't it better to identify precipitates using electron diffraction rather than EDS analysis?
11. Poor quality images of the microstructure shown in Figures 7 and 8.
12. Where sigma phase precipitation was observed and why it is not shown in Fig. 1
13. Reference 10 is identical to 25, they propose to replace one of them with the following publication https://doi.org/10.1016/j.msea.2020.139944.
14. Editorial correction of the file is required.
15. Lack of comprehensive discussion of the results obtained.
Comments on the Quality of English LanguageMinor editing of English language required
Author Response

(The authors gave the same response as above.)

Reviewer 4 Report
Comments and Suggestions for Authors
L13: "after after"
L34: what %
L49-50: this sentence is difficult to read, better rephrase and remove 700°C from brackets
Table 1: %? which %?
Section 3.2 begins with Figure. Please start it better with the text.
Fig. 5f, Fig6 and 7 are difficult to read (element analysis), perhaps as a table?
L206: Secondly/secondly
L222: moreover,NbCrN/ moreover, NbCrN
Author Response

(The authors gave the same response as above.)

Round 2
Reviewer 3 Report
Comments and Suggestions for Authors
In this form the paper can be published
Comments on the Quality of English LanguageThe paper need minor editing of English language.